# Single cell electron collectors for highly efficient wiring-up electronic abiotic/biotic interfaces

Yang-Yang Yu [1,3], Yan-Zhai Wang[1,3], Zhen Fang [1], Yu-Tong Shi[1], Qian-Wen Cheng[1], Yu-Xuan Chen[1], Weidong Shi [2✉] & Yang-Chun Yong [1✉]

By electronically wiring-up living cells with abiotic conductive surfaces, bioelectrochemical systems (BES) harvest energy and synthesize electric-/solar-chemicals with unmatched thermodynamic efficiency. However, the establishment of an efficient electronic interface between living cells and abiotic surfaces is hindered due to the requirement of extremely close contact and high interfacial area, which is quite challenging for cell and material engineering. Herein, we propose a new concept of a single cell electron collector, which is *in-situ* built with an interconnected intact conductive layer on and cross the individual cell membrane. The single cell electron collector forms intimate contact with the cellular electron transfer machinery and maximizes the interfacial area, achieving record-high interfacial electron transfer efficiency and BES performance. Thus, this single cell electron collector provides a superior tool to wire living cells with abiotic surfaces at the single-cell level and adds new dimensions for abiotic/biotic interface engineering.

[1] Biofuels Institute, School of Environment and Safety Engineering, Jiangsu University, 301 Xuefu Road, Zhenjiang 212013, China. [2] School of Chemistry and Chemical Engineering, Jiangsu University, 301 Xuefu Road, Zhenjiang 212013, China. [3] These authors contributed equally: Yang-Yang Yu, Yan-Zhai Wang.
✉email: swd1978@ujs.edu.cn; ycyong@ujs.edu.cn

**W**iring-up living cells with abiotic interface to build up electronic biotic/abiotic interfaces is of great importance for various bioelectronic applications[1,2]. In particular, with the exploration of bidirectional electron exchange between electroactive microorganisms and solid conductive surfaces, bioelectrochemical systems (BES) have been developed and extended as an innovative approach for energy harvesting[3], resource recovery[4], and electric-/solar-chemical production[5–7]. By coupling efficient living cell catalysis and material catalysts, BESs are expected to achieve high thermodynamic efficiency. However, the sluggish biotic/abiotic interfacial electron exchange between the cell and material largely limits the BES performance and practical applications.

Therefore, many efforts have been made to improve the efficiency of biotic/abiotic interface electron exchange[8–12], which is mediated by electroactive cell with transmembrane electron transfer pathways in BES[13,14]. Especially, membrane-bound redox proteins consisting of transmembrane electron transfer conduits have been explored as the most promising pathway to wire up cells with conductive abiotic surfaces[15]. Efficient interfacial electron transfer requires extremely close contact (<14 angstroms)[16,17] between the transmembrane electron transfer conduits and conductive abiotic surface. To date, these abiotic/biotic interfaces used in BES have usually been established following a "top-down" approach, in which an abiotic surface with definite macro- and nanostructure is prefabricated, followed by random attachment of cells to the surface[3]. In this case, the interfacial electron transfer between the individual cell and the conductive abiotic surface is restricted by the following (Fig. 1): (1) only the transmembrane electron transfer conduits with extremely close contact with the abiotic surface are wired up (wired conduits), while there are a large portion of unwired conduits (idle conduits) due to the inherited 3D cell structure; (2) the periplasm-terminated electron transfer conduits (dead conduits) without transmembrane electron transfer ability cannot be wired up. Recently, it was found that nanoparticles scattering aligned on the surface of or inside cells participated in the bacterial transmembrane electron transfer process[18–20], which further inspired the development of carbon-dots feeding strategy to

improve the electron transfer efficiency at the biotic/abiotic interface[21]. However, as the nanoparticles are scattering aligned, electron collection from individual cell is still mainly relies on the "top-down" built bulk electrode, for which the above problems remain unsolved (Supplementary Fig. 1). Thus, it is quite challenging to fully explore the individual cell interfacial electron transfer capability with "top-down" approaches, which calls for a new strategy for interface design.

Here we show a new concept of a "single cell in situ electron collector" that follows the "bottom-up" strategy (Fig. 1). Completely different from the "top-down" design, this "bottom-up" strategy constructs an in situ electron collector (interconnected intact conductive surface) on an individual cell with the aim of maximizing the electron transfer rate and electron recovery at the single-cell level. This "bottom-up" design overcomes the above-mentioned challenges by wiring up the "idle" and "dead" electron conduits at the single-cell level, achieving exceptionally high electron transfer rate and electron recovery efficiency. This proof-of-concept demonstration establishes a promising and new platform for high-performance BES and cellular bioelectronics.

## Results

**Design of single cell electron collectors**. *Shewanella oneidensis* MR-1 (SW) is a model electroactive bacterial species for BES that mainly employ the typical cytochrome-based conduits for transmembrane electron transfer[22,23]. To build up the single cell in situ electron collector, SW cell was chosen and the conventional carbon felt (CF) electrode was selected as the solid conductive surface. Two different kinds of single cell electron collectors were thus proposed (Fig. 1). First, we proposed a bacterial surface anchored electron collector (S collector) that was directly coated on the bacterial outer membrane surface (cell@S), which could guarantee extremely close contact between the S collector and the bacterial transmembrane electron conduits. Moreover, the fully covered and interconnected conductive network formed by the S collector could maximize the electronic interfacial area between the individual cell and the biotic surface. Thus, it is expected that the S collector would wire up more "idle" conduits and improve the interfacial electron transfer (Fig. 1b). Second, to further wire

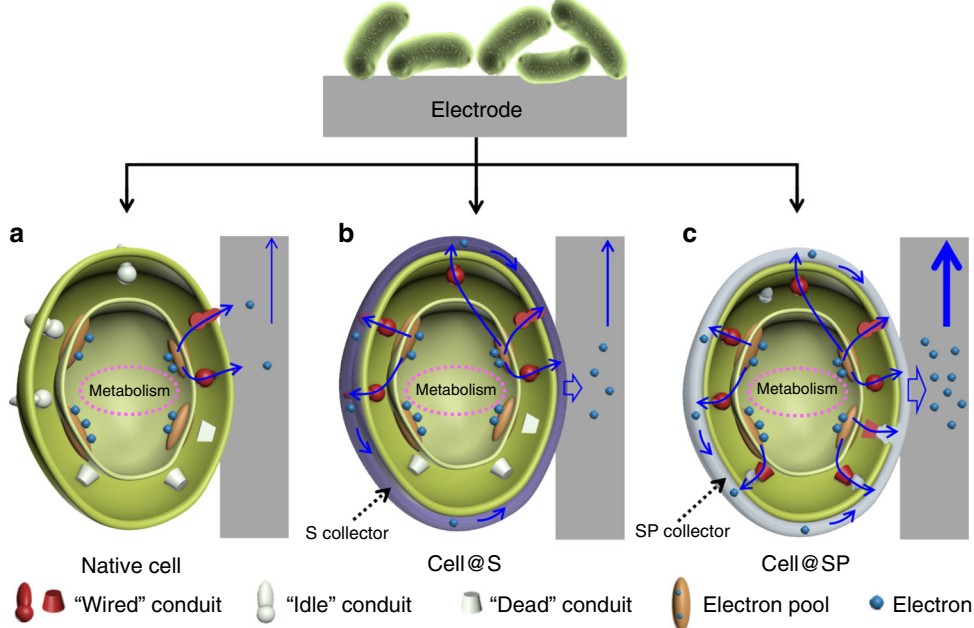

**Fig. 1 Schematic of biointerfacial electron transfer between an electrode and cells. a** Native cell, **b** S collector encapsulated cell (cell@S) and **c** SP collector encapsulated cell (cell@SP).

up the periplasm "dead" electron conduits, the surface and periplasmic single cell in situ electron collector (SP collector) was proposed (cell@SP) (Fig. 1c). The SP collector not only wires up the unwired transmembrane electron conduits (idle conduits) via the surface electron collector, but also bridges the periplasm-terminated electron conduits (dead conduits) with the periplasm-located and/or outer-membrane-embedded electron collector networks. By bridging the "dead" conduits, the SP collector provides extra artificial transmembrane electron conduits for the individual cell and thus further improves the interfacial electron transfer efficiency.

**Assembly of single cell electron collectors.** As polymers can be easily used for individual cell encapsulation[24], we envisioned to construct the S collector with conductive polymers such as polyaniline, polypyrrole, and polydopamine (PDA). PDA is widely used for cell engineering due to its excellent biocompatibility, good conductivity and capability to form uniform nanostructures on versatile surfaces[25,26]. Recently, PDA encapsulation of electroactive biofilms at the population level was achieved[27]. Thus, PDA was selected for S collector fabrication at the single-cell level by in situ polymerization of an interconnected PDA nanoshell on an individual SW cell surface (Fig. 2a, Supplementary Fig. 2a). By simply tuning the polymerization time (the optimum polymerization time is 3 h), a nearly fully covered and interconnected PDA nanoshell (~20–80 nm in thickness) on the cell surface was assembled (Fig. 2b–e and Supplementary Fig. 2b, c). It was observed that the PDA nanoparticles closely contacted the cell outer membrane (Fig. 2e), where the transmembrane electron conduits are usually embedded[28–30]. Thus, the PDA nanoshell was expected to efficiently wire up the transmembrane electron conduits and serve as the S collector

(Fig. 1). Moreover, the PDA nanoshell-encapsulated cells showed high cell viability (99.1 ± 0.1%) (Fig. 2f), suggesting S collector-coated living cell@S cell was successfully assembled.

It was reported that some microbial cells could synthesize various nanoparticles in the cells or on the cell surface through biomineralization[31]. Thus, the SP collector was fabricated by taking the advantage of microbial biomineralization (Fig. 3a). Among various biomineralized nanoparticles (such as FeS, CdSe, Ag and Pd nanoparticles)[20,32–34], FeS nanoparticles were selected for SP collector assembly due to their ease of biosynthesis, high electroactivity and biocompatibility[35–37]. To confine the nanoparticles in the periplasm and on the outer membrane surface, a diffusion-confined biosynthesis strategy was developed (Supplementary Fig. 3a). By controlling the concentration of the $NaS_2O_3$ precursor (0.1 mM), FeS nanoparticles were synthesized that densely anchored on and fully covered the cell surface (Fig. 3b and Supplementary Fig. 4). After removing the cell surface nanoparticles, HAADF-STEM observation showed that nanoparticles were also aligned in the periplasm. More impressively, it was observed that some nanoparticles were embedded in the cell outer membrane (Fig. 3c). Elemental mapping confirmed that these nanoparticles mainly consisted of iron and sulfur (Fig. 3d, e). Moreover, the nanoparticles were characterized as mackinawite FeS by TEM, HRTEM, XRD and XPS analyses (Supplementary Fig. 3b–f). Thus, the FeS network was expected to efficiently wire up the transmembrane electron conduits and bridge the periplasm-terminated electron conduits, which could serve as the SP collector (Fig. 1). The LIVE/DEAD assay also revealed that the FeS nanoparticle encapsulated cells retained high viability (98.3 ± 0.3%) (Fig. 3f), suggesting the SP collector consisting of living cell@SP cell was successfully assembled.

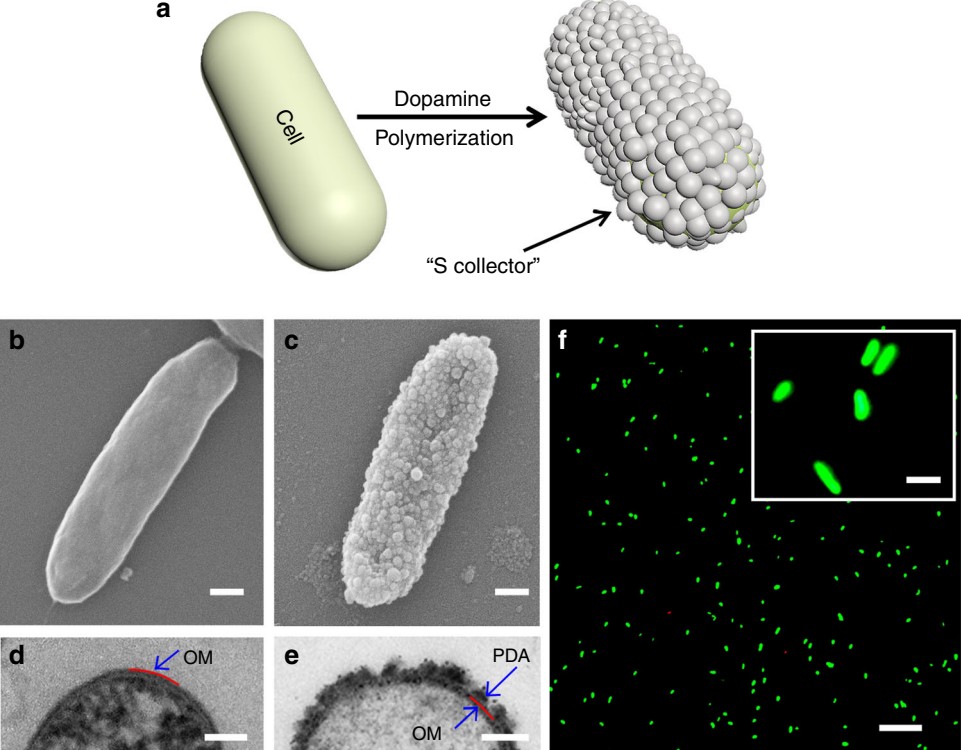

**Fig. 2 Assembly of the S collector on an SW cell. a** Schematic illustration of S collector assembly. SEM images of **b** a native SW cell and **c** an SW@S cell. TEM images of **d** a sliced native SW cell and **e** a sliced SW@S cell. PDA, polydopamine nanoparticle; OM: outer membrane of the SW cell. **f** Fluorescence microscopy image of SW@S cells stained with the LIVE/DEAD assay kit. Green fluorescence indicates living cells; red fluorescence indicates dead cells. Scale bars: **b**, **c** 200 nm; **d**, **e** 100 nm; **f** 20 μm; inset of **f** 2 μm. The inset of **f** shows an enlarged view of stained cells.

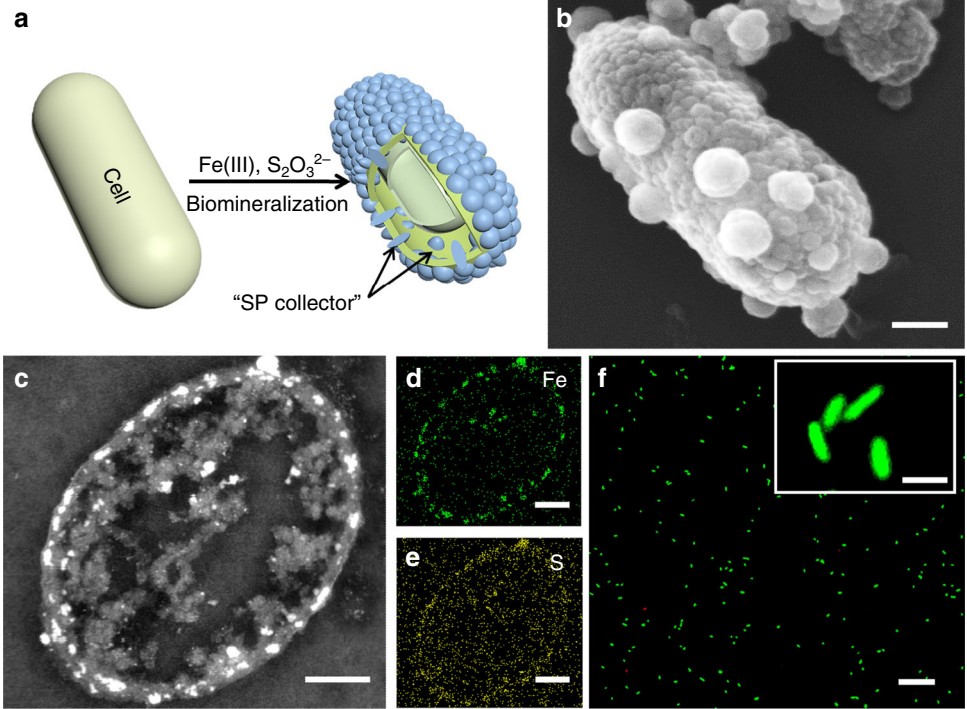

**Fig. 3 Assembly of the SP collector on an SW cell. a** Schematic illustration of the SP collector assembly. **b** SEM image of an SW@SP cell. **c** HAADF-STEM image of a sliced SW@SP cell. **d, e** EDS mapping of Fe or S element for a sliced SW@SP cell of. **f** Fluorescence microscopy image of SW@SP cells stained with a LIVE/DEAD assay kit. Green fluorescence indicates living cells, red fluorescence indicates dead cells. Scale bars: **b, e** 200 nm; **f** 20 μm; inset of **f** 2 μm. The inset of **f** shows an enlarged view of stained cells.

**Wiring cellular conduits with single cell electron collectors.** Next, the function of the single cell electron collectors was validated by evaluating their capability to wire up cellular electron conduits with a bulk solid conductive surface (solid electrode). For the S collector, SW@S cells were loaded on the electrode surface, and the current flow between cells and the electrode was monitored. The SW@S cell delivered a current of approximately 1.9 times that with the native *S. oneidensis* MR-1 cell (native SW) (Fig. 4a, Supplementary Note 1), suggesting that the S collector could electronically wire up the cell with the solid electrode surface. It was further found that disruption of MtrA or MtrC/OmcA (the main components of the MtrC/OmcA-MtrB-MtrA transmembrane electron conduit for SW)[15] greatly inhibited (by over 80%) the current output of the native SW or SW@S cell, implying that the S collector mainly wired the transmembrane electron conduits. Then, the interaction between the S collector and MtrC/OmcA (the outer membrane component of the transmembrane conduit) in the SW@S cell was analyzed in detail by cyclic voltammetry (CV). For native SW cell, the peak pair around −0.219 V attributed to MtrC/OmcA[38,39] was observed in the background-subtracted CV curve (Fig. 4b and Supplementary Fig. 5a). For the SW@S cell, the MtrC/OmcA peaks were extracted from the CV curve of the SW@S cell by subtracting that of the MtrC/OmcA mutant (ΔmtrC/omcA@S). A reversible peak pair around −0.319 V corresponding to the one-electron reduction of flavin-bound MtrC/OmcA[22,40] with a high peak current was observed (Fig. 4b and Supplementary Fig. 5b–d). The higher peak area obtained from the SW@S cell compared with the native SW cell suggested that more MtrC/OmcA conduits were wired up by the S collector. The results substantially proved that the S collector is powerful for wiring up transmembrane electron conduits (Fig. 4c).

Next, the function of the SP collector was verified. As expected, the SW@SP cell delivered an obvious current output, suggesting

its capability to electronically wire up the cell with a solid electrode surface (Fig. 5a, Supplementary Note 1). To further characterize the EET behaviour of the SW@SP cell, CV analyses were performed. Under the non-turnover condition (Fig. 5b), the CV curve of the sterilized SW@SP cells exhibited a well-defined redox pair that was attributed to the redox transformation of sulfur species in FeS[41–43]. For living SW@SP cells or ΔmtrC/omcA@SP cells, an additional anodic peak that might be attributed to the cellular redox components was observed (Fig. 5b). Compared to SW@SP cells, ΔmtrC/omcA@SP cells showed a prominent peak shift and a catalytic current decrease under the turnover condition (Fig. 5c), indicating that the MtrC/OmcA was wired up by the SP collector. Moreover, the maximum current flow reached by an individual cell was substantially inhibited by MtrC/OmcA deletion (ΔmtrC/omcA@SP *vs.* SW@SP) (Fig. 5a). Taken together, these results proved that the SP collector wired up the MtrC/OmcA-MtrB-MtrA transmembrane electron conduits, similar to the S collector.

Strikingly, deletion of MtrC/OmcA only partially inhibited (by 38.1%) the electron flow for the cell with the SP collector (ΔmtrC/omcA@SP) (Fig. 5a), while it nearly fully repressed that of the native SW cell (85.4%) or SW@S cell (83.0%) (Fig. 4a). The results suggested that MtrC/OmcA is not the only transmembrane electron conduit for the SW@SP cell, which is quite different from the native SW cell or SW@S cell. In accordance, there was another anodic peak in addition to that of MtrC/OmcA, and a significant catalytic current after MtrC/OmcA deletion (ΔmtrC/omcA@SP) was observed (Fig. 5b, c). These results suggested that another transmembrane electron transfer conduit in addition to MtrC/OmcA-MtrB-MtrA might be established by the SP collector. It is speculated that periplasm-located and/or membrane-embedded FeS network might bridge the periplasm-terminated "dead" conduits and serve as a newly established artificial transmembrane electron conduit (Fig. 1 and Fig. 3b, c).

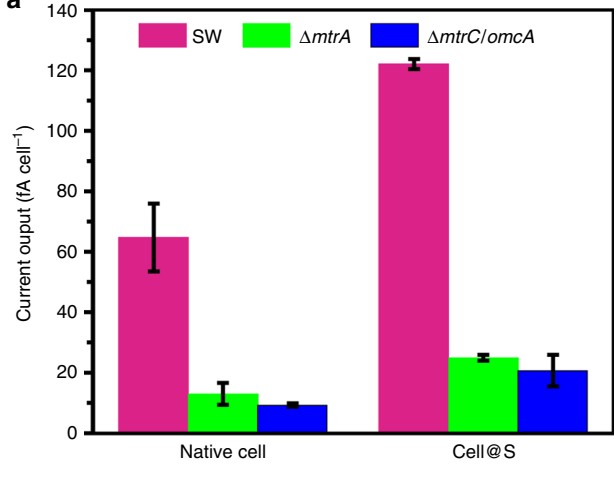

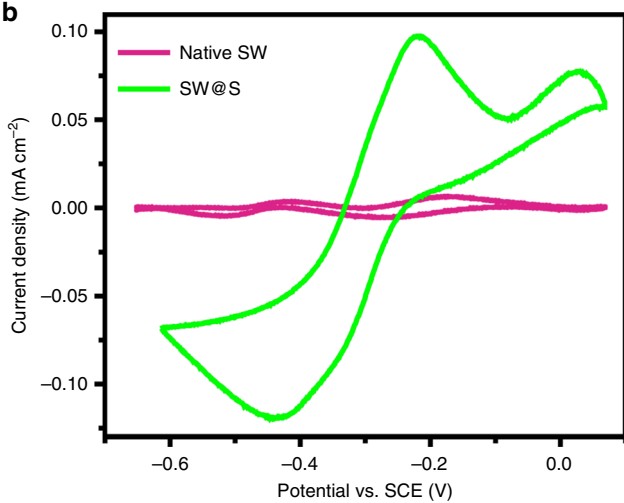

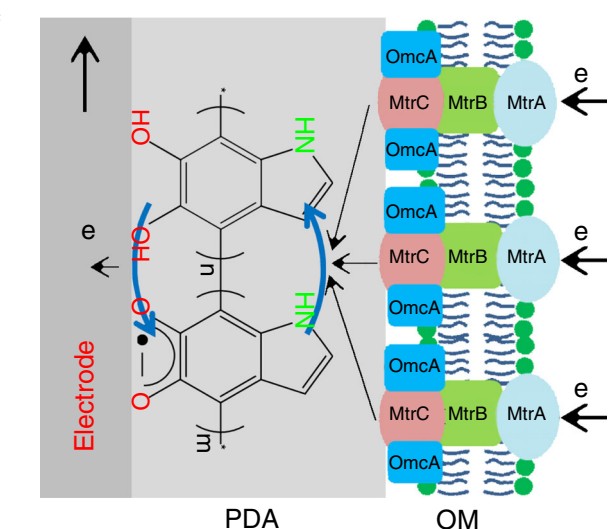

**Fig. 4 Wiring up of cellular conduits with a solid electrode by the S collector. a** Current output of native or S collector encapsulated cells ($n = 3$). Error bars represent standard error (s.e.) determined by three independent experiments. **b** Background-subtracted CV curves of native (magenta) and SW@S (green) cells, the original CV curves are shown in Supplementary Information (Supplementary Fig. 5). **d** Proposed electron transfer pathway for the SW@S cell with an electrode.

The polysulfide reductase PsrABC is the periplasm-terminated electron conduit responsible for FeS biosynthesis[37,44,45]. Thus, it is most likely that PsrABC was bridged by the FeS network and evolved into the FeS-PsrABC hybrid transmembrane electron conduit. Pentachlorophenol (PCP), a competitor of ubiquinone-1, has been recognized as the inhibitor of PsrC (the main component of the PsrABC conduit)[46]. Therefore, the effect of PCP ($2 \, \text{mg L}^{-1}$) on electron flow from the cell to the electrode was investigated. As PsrABC did not participate in the transmembrane electron transfer of the native SW cell, PCP addition only showed a marginal effect (Supplementary Fig. 6). In contrast, the addition of PCP dramatically inhibited the current output of SW@SP cells, with only 13.2% of the current output retained (46 vs. 348 fA cell$^{-1}$) (Fig. 5a). These results indicated that PsrABC made a major contribution to transmembrane electron transfer when the SP collector was assembled, confirming that the SP collector could bridge the periplasm-terminated electron conduits with the extracellular solid conductive surface. Taken together, these results proved that the SP collector could simultaneously wire the "idle" transmembrane electron conduits and bridge the periplasmic-terminated "dead" electron conduits (Fig. 5d), thus offering unique advantages for abiotic/biotic interfacial electron transfer.

To further virtually evaluate the efficiency of this in situ electron collector at the single-cell level, the electron transport between a single cell and an electrode was monitored with a microelectrode assay. Simultaneous cell imaging/tracking and current recording with microelectrode arrays was reported as an effective approach to determine the bacterial electron transport at the single-cell level[47]. By using a microfabricated microelectrode chip, the contact of a single cell with the microelectrode was in situ monitored with a microscope, and the single cell-based stepwise short-circuit current output from the microelectrode was simultaneously detected by an electrochemical workstation (Supplementary Figs. 7, 8)[47]. For the native SW cell, the recorded current steps corresponding to the single cell interaction with the microelectrode showed an average current output of $56 \pm 11$ fA ($n > 50$) (Supplementary Fig. 8b, d). For the SW@SP cell, a substantially higher short-circuit current step for a single cell was recorded, $292 \pm 55$ fA, ($n > 50$) (Supplementary Fig. 8c, e), which was over four times higher than that from a native cell. The single cell current generation monitored with the microelectrode was in good agreement with that estimated from the cell population results ($65 \pm 11$ fA cell$^{-1}$ for native SW and $348 \pm 22$ fA cell$^{-1}$ for SW@SP) (Figs. 4a and 5a). These results further confirmed that the single cell electron collector could efficiently improve the abiotic/biotic interfacial electron transfer efficiency at the single-cell level.

**High-performance BES enabled by single cell electron collectors.** Furthermore, the performance of the single cell electron collectors was evaluated with a typical BES system of microbial fuel cells (MFCs). As expected, the MFCs with the SW@S cell ($\sim 0.123 \, \text{mA cm}^{-2}$) or SW@SP cell ($\sim 0.152 \, \text{mA cm}^{-2}$) exhibited a much higher maximum current output than that with the native SW cell ($\sim 0.079 \, \text{mA cm}^{-2}$) (Supplementary Fig. 9). The MFCs without cells or MFCs with dead SW@S or SW@SP cells did not deliver a significant current output, while the MFCs with living cells could produce a high current output (Supplementary Fig. 10, Supplementary Note 2), indicating that the current was derived from the cells in the MFC. Moreover, direct modification of the electrode with nanomaterials of S or SP layer only showed marginal effect on the current output of native SW cells (Supplementary Fig. 10b). Strikingly, the MFCs with the SW@S or SW@SP cell showed a much longer discharge lifetime than those with the native SW cell (Supplementary Fig. 9). The Coulombic

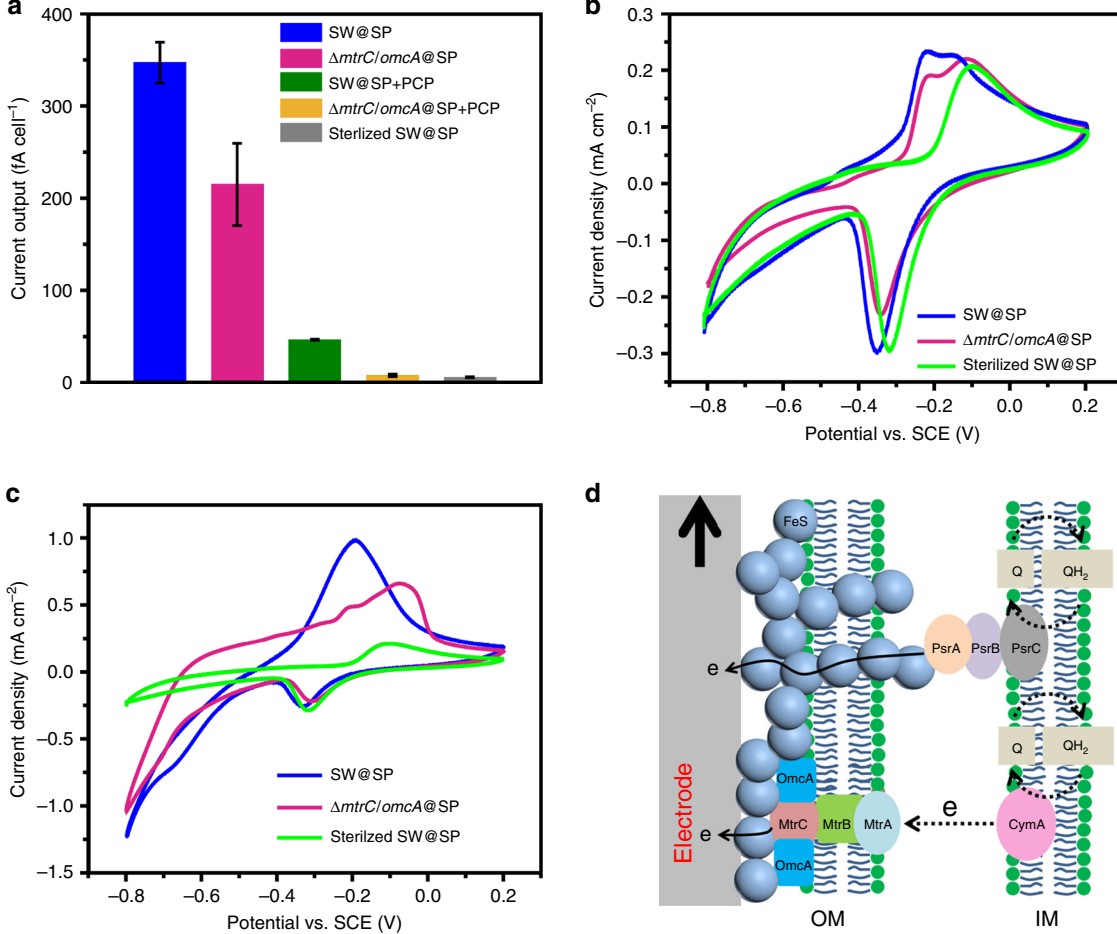

**Fig. 5 Wiring up of cellular conduits with a solid electrode by the SP collector. a** Current output of an cell@SP cell under different conditions ($n = 3$). PCP (2 mg L$^{-1}$, PsrC inhibitor) was added to SW@SP or $\Delta mtrC/omcA$@SP cells at 2 h before electrochemical test. Error bars represent standard error (s.e.) determined by three independent experiments. **b**, **c** CV curves of SW@SP cells under non-turnover and turnover conditions with a scanning rate of 1 mV s$^{-1}$. **d** Proposed electron transfer pathway for the SW@SP cell with an electrode.

**Table 1 Performance of BES with native cells or cells with different single cell electron collectors.**

| Biointerface setup | P$_{max}$ (W m$^{-2}$) | Max. e$_f$ flux (electrons s$^{-1}$ cell$^{-1}$)[a] | CE (%) |
|---|---|---|---|
| Native SW | 0.20 ± 0.01 | 0.77 × 10$^6$ | 14.7 |
| SW@S | 0.66 ± 0.02 | 1.05 × 10$^6$ | 25.1 |
| SW@SP | 3.21 ± 0.27 | 2.50 × 10$^6$ | 86.9 |

[a]The maximum electron flux per cell was estimated from the maximum current output reached in polarization curves (Fig. 6b).

efficiency (CE) was increased from 14.7% (native SW cell) to 25.1% (SW@S cell) by S collector assembly (Table 1). More impressively, the SP collector increased the CE to an extremely high level of 86.9% (SW@SP cell), which is the highest value for the model electroactive bacteria *S. oneidensis* MR-1 obtained in MFCs (Table 1 and Supplementary Table 1). Electrochemical impedance spectroscopy (EIS) analysis revealed that the S collector or SP collector reduced the interfacial charge transfer resistance by approximately 2.5 and 25 times, respectively (Fig. 6a). These results confirmed that these single cell electron collectors enabled more efficient interfacial electron transfer between cells and the extracellular solid conductive surface.

The polarization curves and power output curves were measured to further quantitatively evaluate the performance of the single cell electron collectors. As shown in Fig. 6b, the slopes of the polarization curves obtained from the MFCs with SW@S cells or SW@SP cells were much smaller than that with native cells, implying a smaller internal resistance, in good agreement with the EIS analysis (Fig. 6a). In accordance, the power output of the MFCs was greatly improved by these single cell electron collectors (Fig. 6c). Impressively, the maximum power density of the MFCs with SW@SP cells achieved 3.21 W m$^{-2}$, which was 14.5 times higher than that of the native SW cells (0.207 W m$^{-2}$) and the highest output recorded with this model strain (Fig. 6c, Table 1 and Supplementary Table 1)[3].

Furthermore, it was found that the estimated maximum electron transfer rate from a cell to the solid electrode was increased from 0.77 × 10$^6$ electrons s$^{-1}$ cell$^{-1}$ (native SW) to ~1.05 × 10$^6$ electrons s$^{-1}$ cell$^{-1}$ (SW@S) or ~2.5 × 10$^6$ electrons s$^{-1}$ cell$^{-1}$ (SW@SP) (Table 1). More surprisingly, the single cell electron transfer rate from the SW@SP cell to the solid conductive surface nearly reached that of cell to soluble electron acceptors (≤2.8 × 10$^6$ s$^{-1}$ cell$^{-1}$) (Supplementary Table 2). The results substantiated that the single cell electron collectors improved the abiotic/biotic interfacial electron transfer efficiency and BES performance at the fundamental single-cell level, which is promising for breaking the limit of abiotic/biotic interfacial electron transfer.

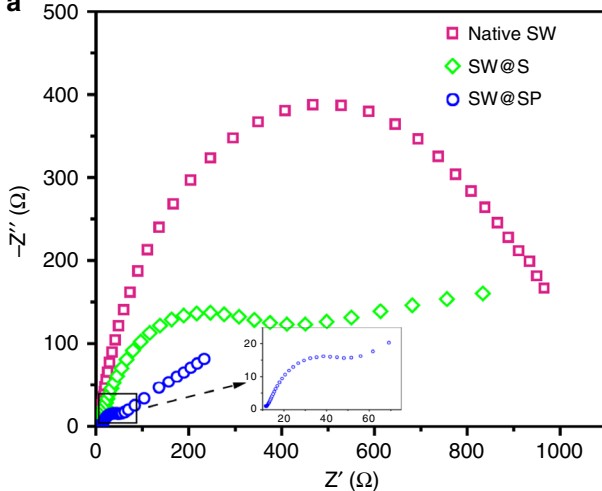

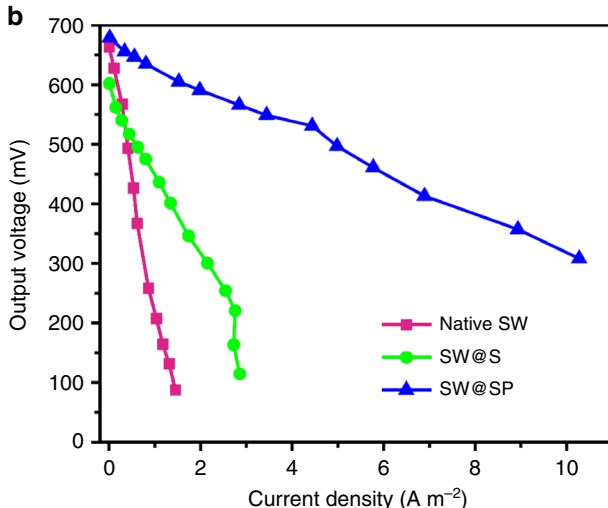

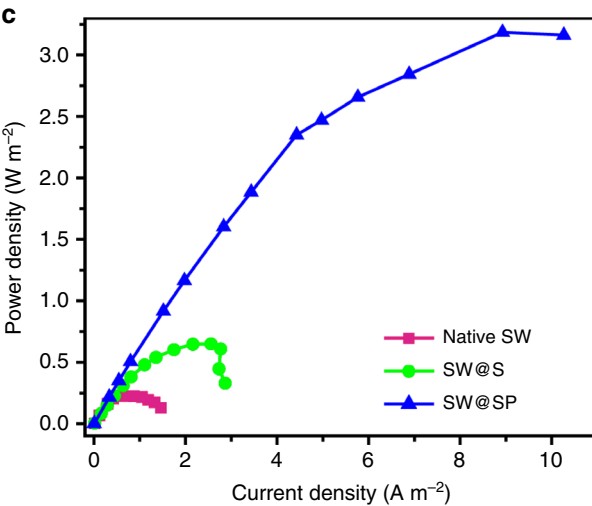

**Fig. 6 Performance of BES inoculated with different cells. a–c** EIS spectra, polarization curves and power output curves of MFCs with different cells.

The individual cell electron transfer efficiency (rate and recovery) is the most essential parameter that determines the performance of living cell electronics. As the single cell in situ electron collectors developed here can wire up more transmembrane electron conduits and establish new

artificial transmembrane electron conduits, it is reasonable to expect a higher electron transfer efficiency. Nevertheless, it is still surprising that the SP collector can enable a record-high single cell electron transfer rate to a solid electrode that is nearly equal to that obtained with soluble electron acceptors (Supplementary Table 2). It is well recognized that electron transfer between cells and soluble electron acceptors could occur in the periplasm reductases or electron pool. In contrast, the electron transfer from a cell to a solid conductive surface should be relayed by tandem transmembrane electron conduits[13]. Thus, it is suggested that the relay process might limit the electron transfer rate to a certain extent[48]. Therefore, although the S collector wired nearly all the outer membrane conduits, the electron transfer rate was still limited by the relayed transmembrane electron transfer process, which resulted in a lower electron transfer rate than that with a soluble electron acceptor (Supplementary Table 2). Howveer, the SP collector could directly wire the periplasm conduits to the extracellular solid conductive surface in a one-step conductive FeS network-mediated manner (Fig. 5d), which might overcome the limitation of multistep relayed transmembrane electron transfer and result in fast electron transfer.

It is well known that some bacterial species also employ freely diffusive electron shuttle-mediated electron transfer for abiotic/biotic interfacial electron exchange. Apart from the transmembrane redox proteins, it was reported that flavins secreted by *S. oneidensis* MR-1 also played roles on interfacial electron transfer[49]. However, recent reports suggested that flavins might mainly bind with the outer membrane cytochromes rather than serving as diffusive electron shuttles depending on the operation conditions[22,50,51]. In this case, flavins might be considered as cofactor and a part of the redox protein-mediated electron transfer pathway[22], where the interfacial electron transfer still occurs in a direct-contact-based manner, similar to that of pure redox protein-mediated electron transfer (Fig. 1a). In this study, cytochrome-bound flavin was also detected (Fig. 4b), while disruption and inhibition of redox proteins (Mtr and Psr pathways) resulted in current decrease of over 97% for SW@SP (Fig. 5a). These results indicated that the redox protein-mediated pathways mainly contributed to the interfacial electron transfer of *S. oneidensis* MR-1 under current conditions, while the contribution of freely diffusive flavins-mediated electron transfer might be negligible. Therefore, the single cell in situ electron collector mainly wired the contact-based redox protein-mediated pathways of *S. oneidensis* MR-1 to improve the interfacial electron transfer efficiency. However, it is still interesting to envision that this in situ electron collector might be useful for shortening the electron shuttling distance (shifting the electron shuttling route from "periplasm/cytoplasm-electrode surface" to "periplasm/cytoplasm-in situ electron collector") of the freely diffusive electron shuttles, which would also be beneficial to improving the efficiency of freely diffusive electron shuttle-mediated interfacial electron transfer.

In addition to the individual cell activity, the biofilm conductivity also determines the system performance of bioelectronics[52,53]. In a biofilm, the electrons released from far-away cells are transferred to the solid conductive surface via an electron mediator or conductive pili relayed process[49,54]. Thus, the construction of an artificial conductive network between cells in biofilm would be favourable for substantially improving the biofilm conductivity and system performance[55,56]. The S collector and SP collector encapsulation modified the individual cell into a conductive "micro-capsule", which could eventually assemble into an interconnected conductive biofilm network. This conductive network should facilitate electron transfer across the whole biofilm. Taken together, the single cell electron collector not only improved the electron transfer efficiency of the

individual cell but also facilitated electron transfer across the biofilm. Thus, the power output obtained from the MFC with the SW@SP cell reached 3.21 W m$^{-2}$, which is the highest power density achieved with this model electroactive bacteria (Supplementary Table 1). It is worth to note that this power density obtained here just based on the proof-of-concept demonstration without systematic optimization. Comprehensive investigation of the effect of the thickness, particle size, and conductivity of the single cell electron collector layer on the electron collection efficiency and systematic optimization would further improve the electron transfer efficiency as well as the BES performance.

**Stability and versatility of single cell electron collectors**. Inheritance of artificial nano-modification to the following generation of cells remains a challenge for the fabrication of biohybrid systems[1,57]. However, it was reported that nanoencapsulation of individual cells might arrest cell growth to some extent[24]. The growth curves of native SW, SW@S and SW@SP cells showed that the in situ electron collector encapsulation suppressed the cell division activities and prolonged the lag phase under anaerobic cultivation conditions (the growth of SW@SP or SW@S cells was arrested for ~4 h or 18 h, respectively) (Supplementary Fig. 11). Moreover, it was found that the SW@SP biofilm on the electrode showed no significant cell growth for over 24 h (from 20 h upon maturation of the biofilm to 48 h) and maintained stable interfacial electron transfer under electrode respiration cultivation conditions (from 20 to 60 h) (Supplementary Fig. 12a, b). The growth arrest property would be advantageous for short- to mid-term applications (hours to a day), as it might provide a stable interface without cell population disturbance. However, for extremely long-term applications, cell growth is inevitable and might partially disrupt the intact single cell electron collector, produce extracellular polymer substance (EPS) (Supplementary Fig. 12c) that partially dampen the biofilm conductivity, which might be unfavourable for maintaining the high interfacial electron transfer efficiency. Thus, further upgrading of this in situ electron collector layer with a dynamic/self-repairing nanoshell, or development of a genetic-encoding conductive nanoshell with synthetic biology is expected, which would enable the establishment of a next-generation intelligent and inherited single cell electron collector to further extend its applications.

Moreover, the single cell in situ electron collector design developed here could be easily adapted to versatile systems. Recently, many approaches have been developed for individual cell encapsulation[57,58], which provides great potential to expand the tool box for S collector assembly. With these state-of-art approaches, it is envisioned to install S collectors for different bacteria with more elaborated architectures and tailored functions. Moreover, the recipe for biomineralization has also been dramatically extended to nearly cover all metal elements[59]. Genetic engineering would further endow bacteria with an unprecedented capability to produce versatile conductive/functional nanoparticles with fine tuning of the morphology and localization. With the broad recipe and control possibilities, the SP collector could be assembled with high flexibility for various bioelectronic interfaces. In addition, the single cell electron collector could also serve as an electron distributor to efficiently distribute the electrons from the solid conductive surface into individual cells (Supplementary Fig. 13), which would extend its broad applications to electricity- or solar-to-chemical production.

## Discussion

In summary, we demonstrate a new concept of single cell electron collector for highly efficient wiring-up electronic abiotic/biotic interfaces. As a proof-of-concept, S collector that directly coated on the bacterial outer membrane surface is assembled by in situ polymerization of dopamine on the *S. oneidensis* MR-1 cells, which wires up the "idle" bacterial transmembrane electron conduits. Meanwhile, SP collector that assembled by tuning the microbial biomineralization of FeS nanoparticles, simultaneously wires up the bacterial "dead" periplasmic and "idle" transmembrane electron conduits. As a result, the electron transfer rate from an individual cell to a solid electrode is promoted by using single cell electron collectors ($\sim 2.5 \times 10^6$ electrons s$^{-1}$ cell$^{-1}$) to that of cell with soluble electron acceptors. Furthermore, the maximum power output of 3.21 W m$^{-2}$ is achieved in the MFC that inoculated with SW@SP cells, which is the highest record with this model strain. Hence, our work opens a new avenue to improve the abiotic/biotic interfacial electron transfer efficiency at the fundamental single-cell level, which is promising for breaking the limit of abiotic/biotic interfacial electron transfer and extending its broad applications for electricity- or solar-to-chemical production.

## Methods

**Bacterial strain and cultivation**. The SW strain and the corresponding mutants Δ*mtrA* and Δ*mtrC/omcA* were aerobically cultivated in Luria-Bertani (LB) broth at 30 °C under shaking (200 rpm)[56]. When the OD$_{600}$ reached 3.0 (about 16 h), the cells were harvested by centrifugation (5000 rpm, 5 min) for further use. For all electrochemical tests, the harvested cells were resuspended in a nitrogen gas purged (30 min) medium, and the following operations were all performed under anaerobic conditions.

**Assembly of the S collector on cells**. The harvested bacterial cells were resuspended in Tris-HCl buffer (10 mM, pH = 8.5) to an optical density of OD$_{600}$ = 4. Then, dopamine hydrochloride (4 mg mL$^{-1}$) was added to the cell suspension. The cell/dopamine mixture was aerobically incubated with shaking (200 rpm) for in situ polymerization of PDA on the cell surface for 0.5~4 h at 30 °C[26]. After polymerization, the cell pellet was collected by centrifugation (5000 rpm, 5 min) and washed two times with nitrogen gas purged distilled water. A cell that was fully coated with a PDA nanoshell was designated as an SW@S cell.

**Assembly of the SP collector on cells**. The FeS based SP collector was assembled under anaerobic incubation. Bacterial cells (OD$_{600}$ = 0.1) were suspended in M9 salt supplemented with yeast extract (0.5 g L$^{-1}$), peptone (0.25 g L$^{-1}$), sodium lactate (18 mM), FeCl$_3$ (0.1 mM) and Na$_2$S$_2$O$_3$ (0.1 mM). The mixture was bubbled with nitrogen (20 min), sealed and incubated for 12 h with shaking (200 rpm). After synthesis, the cell pellet was collected by centrifugation (5000 rpm, 5 min) and washed two times with nitrogen gas purged distilled water. A cell with a FeS nanoshell was designated as an SW@SP cell.

**Characterization of electron collector assembled cells**. Microscopy imaging was conducted as follows: (1) SEM: diluted cell sample was fixed with glutaraldehyde 2.5% (v/v) containing PBS buffer on a silica wafer for 4 h, washed with saline solution three times, sequentially dehydrated with gradient ethanol/water solutions, and finally observed with field emission scanning electron microscope (JSM-7800F, JEOL, Japan). (2) Fluorescence microscopy: the cell suspension was centrifuged (5000 rpm, 10 min) and the pellet was collected. Then, the pellet was resuspended in 0.85% NaCl. Next, 3 μL of dye mixture (PI/SYTO 9, Live/Dead Backlight Bacterial Viability Kit L7012) was added for each mL of the bacterial cell suspension and incubated for 15 min in dark at room temperature. After that, 5 μL of the stained cell suspension was pipetted on a microscope slide and observed with a fluorescence microscope (MF30, Guangzhou Mshot Photoelectric Technology CO., LTD., China). For cell viability quantification, statistical analysis of the ratio of live/dead cells from multiple images was conducted. (3) TEM: cells were fixed with glutaraldehyde (2.5%, v/v) (for native SW and SW@SP cells) or glutaraldehyde (2.5%, v/v) with paraformaldehyde (2%, v/v) (for SW@S cells)[60], stained with osmic acid and wrapped in resin. After that, the samples were further sliced and observed with TEM or HAADF-STEM-EDS (Tecnai G$^2$ F20 S-Twin, FEI, USA).

FeS samples were harvested by lysing SW@SP cells with sonication, centrifugation, washing and drying in an anaerobic workstation (Ruskinn Bugbox M, Baker, UK). The morphology and composition were characterized with TEM, HRTEM (FEI Tecnai F30, FEI, USA), X-ray photoelectron spectroscopy (Escalab 250Xi XPS, Thermo Scientific, USA) and X-ray diffractometer (D8 Advance, Bruker, German).

**Bioelectronic device setup**. Dual-chamber MFC (inner size of 2 × 4 × 4 cm) separated by a Nafion117 membrane (3 × 3 cm, DuPont, USA) were used. A CF electrode (1 × 2 × 0.5 cm) was used as the anodic electrode[61]. M9 salt medium with Wolfe mineral and Wolfe vitamin was used as the anode medium[62], and sodium lactate (18 mM) was supplemented as the sole carbon source. Native SW cells or SW@S cells were suspended in the anode medium to an OD$_{600}$ = 1.0, and purged

with nitrogen gas for 30 min to remove the dissolved oxygen. Then, the cell suspension was filled into the anode chamber of the MFC in an anaerobic workstation (Ruskinn Bugbox M, Baker, UK) and tightly sealed to maintain anaerobic conditions during operation. For SW@SP cells, the SP layer was in situ assembled in the MFC anode chamber. In brief, $FeCl_3$ (5 mM) and $Na_2S_2O_3$ (5 mM) were added into the anode chamber of a native SW cell-inoculated MFC (all operations were performed in an anaerobic workstation and the MFC chamber was tightly sealed to maintain anaerobic conditions). The SW@SP cells were successfully assembled after 15 h incubation. A CF electrode (2 × 3 cm) was used as the cathodic electrode, and ferricyanide solution (50 mM) was used as the catholyte. The assembled MFC was connected with 2 kΩ resistors and discharged at 25 °C. At least three independent MFCs were prepared for each test.

**Electrochemical characterization.** The output voltage was recorded by a 15B digital multimeter (Fluke, USA) with an MPS-010602 data collector (Qicuang, Beijing, China). The polarization curves were measured by changing the external resistance. The current density at the maximum power output[63] was adopted for comparing the current output of SW cells (with or without a single cell electron collector, deletion of MtrC/OmcA, or addition of PCP). The maximum current density from the polarization curve was used for the calculation of the maximum electron flux of a single cell. CV analyses were conducted by using a CHI660E electrochemical workstation (CHI, Shanghai, China) with a platinum wire counter electrode and a saturated calomel electrode reference electrode. EIS analyses were conducted with an Interface 1000 potentiostat (Gamry, USA). All the current density and power density were normalized to anode projected area unless otherwise indicated.

**Electrode biofilm staining and observation.** For the observation of the biofilm on an electrode, a piece of electrode from an MFC was collected and rinsed three times with 0.85% NaCl. Then the electrode piece was immersed into 1 mL of SYTO 9 solution (30 μM) for 15 min in the dark at room temperature. Next, the electrode piece was rinsed two times with 0.85% NaCl. After that, it was subjected to rhodamine labelled concanavalin A (rhodamine-ConA, 10 μg/mL) staining for 15 min in the dark at room temperature. Finally, the electrode piece was rinsed two times with 0.85% NaCl and directly observed with a fluorescence microscope (MF30, Guangzhou Mshot Photoelectric Technology CO., LTD., China).

**Microelectrode chip fabrication and measurements.** Microelectrode chip fabrication and electrochemical measurements were performed according to a previous report with minor modifications[47]. The microelectrode design is shown in Supplementary Fig. 7a. In brief, glass substrates were cleaned, and a microelectrode array (12 Au finger electrodes (4-μm wide for each electrode) and an Ag reference electrode (40 μm in width, 500 μm in length)) was deposited on the slide by electron-beam evaporation. After lift-off, an SU-8 layer with a thickness of ~50 μm was deposited. Photolithography was then used to define the flow chamber (200 μm in width). The position of the flow chamber was precisely controlled to expose the same length of the Au finger electrodes of 18 μm. Finally, a glass coverslip (0.1 mm) was covered on the SU-8 layer to form a closed flow chamber with defined inlets and outlet. For cell suspension loading, the operations were conducted in an anaerobic workstation. Then, the inlets and outlet of the microelectrode chip were tightly sealed to maintain anaerobic condition during electrochemical measurements.

For electrochemical measurements with this microelectrode chip, a CHI660E electrochemical workstation (CHI, Shanghai, China) was used. The microelectrode chip and microscope were housed in a Faraday cage to reduce the noise signal. The short-circuit current was recorded with a reference/counter electrode grounded as reported[47]. In situ optical imaging of cells in the microelectrode chip was carried out by using a microscope (MF30, Guangzhou Mshot Photoelectric Technology CO., LTD., China) with a ×100 oil-immersion lens. A bandpass filter (400–700 nm) was also used to block UV and IR light to avoid possible detrimental effects on the cells.

**Reporting summary.** Further information on research design is available in the Nature Research Reporting Summary linked to this article.

## Data availability

The authors declare that all data supporting the findings of this study are available from the corresponding authors on request.

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

## Acknowledgements

This work was partially supported by the National Natural Science Foundation of China (51708254, 31870112, 21878129), Fok Ying-Tong Education Foundation (Grant no., 161074), Natural Science Foundation of Jiangsu Province (BK20170545), Zhenjiang Key Laboratory of Advanced Sensing Materials and Devices (No.SS2018001), Open Funding Project of State Key Laboratory of Microbial Metabolism (MMLKF18-03), Senior Talent Funded Projects of Jiangsu University (17JDG016) and a project funded by the Priority Program Development of Jiangsu Higher Education Institutions.

## Author contributions

Y.Y.Y. and Y.C.Y. conceived the project and designed the experiments. Y.T.S. and Y.Z.W. carried out the BES analysis. Q.W.C., Y.Y.Y. and W.D.S. conducted the SW@SP assembly and characterization. Y.X.C. and Y.Z.W. conducted SW@S assembly and characterization. Y.Y.Y., Y.Z.W. and Y.C.Y. conducted microelectrode assay. Y.Y.Y., Z.F., W.D.S. and Y.C.Y. analyzed the data and wrote the manuscript. Y.C.Y. supervised this research. All authors read and approved the manuscript.

## Competing interests

The authors declare no competing interests.
