## [Peer Review File · Nature Communications]

Reviewers' comments:

Reviewer #1 (Remarks to the Author):

The aim of this study was to propose a new concept of single cell electron collector, which was in-situ built up with an intact conductive layer on and cross the individual cell membrane. The authors used this method to achieve record-high interfacial electron transfer efficiency and bioelectrochemical systems performance. However, the authors fail to distinguish this work from previous publications. Therefore, I recommend a major revision of the manuscript as indicated.

1. How to identify the interfacial electron transfer efficiency of single cell? In BES, please provide the CLSM or SEM of the bacteria on the carbon electrode. Was the biofilms formed? The interfacial electron transfer of single cell or biofilms is different (Torres CI et al., *Fems Microbiol Rev*, 2010, 34: 3-17). Electrochemical workstation and in-situ micro imaging and micro fabrication technology might help to conduct this experiment (please refer to Jiang et al, *P Natl Acad Sci USA*, 2010, 107 (39), 16806-16810; Jiang et al., *Nat Commun* 2013, 4, e2751).

2. As widely reported, many research groups have used the similar ways to achieve record-high interfacial electron transfer efficiency. For instance, Du et al., *Environ. Sci. Technol. Lett.* 2017, 4 (8), 345-349, intact conductive layer on the individual cell membrane; Wu X et al., *Angew. Chem. Int. Edit.*, 2011, 50 (2), 427-430, intact conductive layer cross the individual cell membrane. The authors should clearly elaborate the significant improvement of this work.

3. Electron shuttle is well known to efficiently mediate microbial EET for *Shewanella oneidensis* MR-1. This point should be added into the introduction. Also, more discussion about this issue should also be added. Was this intact conductive layer a type of solid electron shuttle?

4. In order to exclude other reasons to improve the interfacial electron transfer efficiency, the interfacial electron transfer efficiency between the native *Shewanella oneidensis* MR-1 and the electrode modified with PDA or FeS₂ should be examined.

5. Considering that an intact conductive layer was on and cross the individual cell membrane, could the interfacial electron transfer efficiency maintain at a high level during the bacterial cultivation period?

Reviewer #2 (Remarks to the Author):

This submitted work attempts to maximise the interfacial electron transfer and bioelectrochemical systems (BES) performance between living electroactive cells and conductive abiotic surfaces. This engineering attempt is termed as wiring to enhance intercellular connection because close surface contact is essential for maximal electrical output. The author justified the work in reference to current methods that lack in improving the cellular properties for efficient electron transfer, but only focusing on the properties of the conductive abiotic surfaces. For electron transfer improvement, the work targeted two parts of the model electroactive organism, *Shewanella oneidensis* MR-1: (1) "idling" transmembrane electron conduits and (2) "dead" periplasmic electron conduits, both of which are not in contact with electrode surface. To engage these inactive electron conduits, the electroactive bacteria were encapsulated with either polydopamine (PDA) or anchoring FeS nanoparticles in the periplasmic and on outer membrane surface, to act as electron collectors that was shown to increase the BES performance by 1.4- to 3.2-fold, respectively, in comparison to the intact cells. This work demonstrates a proof-of-concept for optimizing BES performance by modifying cell surface chemistry towards contact enhancement with a conductive surface, of which is an attractive strategy to be exploited for potential bioelectronic applications. This is a very interesting paper and a good read.

Overall, the results are quite impressive but there are some concerns that need to be resolved:

1. Introduction of PDA and FeS as nanomaterials were rather abrupt. How did the author decide on the materials to be used as electron collectors? Was there a panel of materials compared or at least considered to give the desired pooling of electrons like PDA?
2. In reference to Figs 2 and 3, the TEM image of native SW surface (2d) displayed homogenous distribution of its cytoplasmic content. However, the TEM images of SW@S (2e & 2f) and SW@SP (3c) displayed significant clusters of its cytoplasmic content. It seems to concentrate at the outer membrane in SW@S, while it clumps in SW@SP. Is there any explanation on these observations?
3. The authors must briefly describe the Live/dead staining method, for example, how long it was incubated post staining, dyes used and excitation/emission wavelengths of the respective dyes. The live/dead stained images, (Fig 2g and Fig 3f), shows all cells to be live. Just wondering how come there is not even a single dead/membrane compromised cell. It would be good to quantify the amount of live/dead cells, post encapsulation (S and SP), from multiple images and present the percentage of live and dead cells.
4. Live/Dead stains gave a reasonable idea on their viability but do these cell surface modifications have any effect on their growth? The surface modifications are not of genetic engineering nature. Hence, it is not a feature that can be inherited for the following generations. Encapsulation/FeS anchoring would be diluted while cells grow. This aspect needs to be explained/discussed.
5. The work discussed on the possibility of networking as an electroconductive biofilm. Technically, biofilms form through their embodiment in a self-secreted biological matrix (extracellular polymeric substances, EPS). In relation to my concern #4, if the surface modification is not inherited, the mentioned biofilm could possibly consist of both surface modified and native cells as well as large amount of EPS. In this case, the discussed "construction of artificial conductive network" (page 13, line 252) may still be possible but dampened with presence of unmodified cells and EPS. It would be good if the authors could provide some biofilm (including EPS) images of S or SP capsulated cells. The effect of EPS should be discussed.
6. The process of S collector assembly has been explained on page 14 (line 289- 294). Process needs more details for the reproducibility. How much time was required for polydopamine coating on the surface of bacteria? How thick is the PDA layer and how does the thickness affect the EET performance? Similarly, since S collector and SP collector are based on nanostructures, how does the size of these nanostructures affect the EET performance?
7. The effect of PCP on the current output has been shown on Page 9 (line 181-187). However, authors did not mention the concentration of PCP used in the methods section (page 17, line 338-340). The concentration of inhibitor is critical as we do not know if the same concentration of inhibitor was used for all 3 types of cells or not.
8. Bioelectronic setup (page 16, line 322-333) needs more details. The authors mentioned that in-situ assembly of SP collector was done by adding precursors in same ratio (line 328-330). However, the time required for in-situ assembly should also be mentioned here. Was it the same as that required for general process of SP collector assembly on the cells?
9. It was mentioned earlier (page 15, line 296) that anaerobic conditions are necessary for FeS based SP collector assembly. How anaerobic conditions were maintained during in-situ assembly (page 16, line 322-333)?
10. Figure legend 2g and 3f: Please change "strained" to "stained".

11. Supplementary Figure S2: HRTEM image of FeS nanoparticles collected from SW@SP, not clear. Please replace the image.

Authors' Responses to the Reviewers' Comments

(Manuscript ID: NCOMMS-19-42215)

We are very grateful for all the comments and suggestions from the editor and all the reviewers. These comments and suggestions are very important and valuable to improve the quality and readability of this paper. After the revision according to the comments and suggestions, the quality of the manuscript is substantially improved with clearly defined novelty and more convincing data (7 additional figures and more discussion are provided). Detailed revisions and responses to address your comments are presented as below.

Reviewer #1:

The aim of this study was to propose a new concept of single cell electron collector, which was in-situ built up with an intact conductive layer on and cross the individual cell membrane. The authors used this method to achieve record-high interfacial electron transfer efficiency and bioelectrochemical systems performance.

REPLY: We highly appreciate the reviewer's positive comments on this work.

However, the authors fail to distinguish this work from previous publications. Therefore, I recommend a major revision of the manuscript as indicated.

REPLY: Thanks very much for this comment and suggestion. As shown in the following responses (pls refer to the details of reply to question 2), we carefully revised the manuscript and distinguished our work with the previous publications, which makes the novelty of this work was fully defined. We really appreciate the reviewer's comment which is of great help to improve the quality of this manuscript.

ACTIONS: We introduced the recent progress on cell modification with nanomaterials in the introduction section (page 4, line 62-69) and provided a supplementary figure (Supplementary Fig. S1) to illustrate the main ideas of these

previous publications, which could be easily distinguished from the concept of the current work.

1. How to identify the interfacial electron transfer efficiency of single cell? In BES, please provide the CLSM or SEM of the bacteria on the carbon electrode. Was the biofilms formed? The interfacial electron transfer of single cell or biofilms is different (Torres CI et al., Fems Microbiol Rev, 2010, 34: 3-17). Electrochemical workstation and in-situ micro imaging and micro fabrication technology might help to conduct this experiment (please refer to Jiang et al, P Natl Acad Sci USA, 2010, 107 (39), 16806-16810; Jiang et al., Nat Commun 2013, 4, e2751).

REPLY: Thanks for this important suggestion. We agree that microelectrode chip as suggested is more appropriate to measure the interfacial electron transfer of single cell. So, in the revised manuscript, we further identified the interfacial electron transfer efficiency of single cell with microelectrode chip measurement (Supplementary Fig.7-8). The fluorescence image of biofilm on the electrode was provided (Supplementary Fig. 12c). By using this *in-situ* micro imaging and microelectrode technology as suggested by the reviewer, the electron transfer efficiency of single cell reported here is more convincing and the quality of this manuscript is greatly improved. We really appreciate the suggestion from the reviewer.

ACTIONS: We conducted the electrochemical measurement with *in-situ* micro imaging and microelectrode technology. The results are presented in Supplementary Fig. 8 and described in the results section of the revised manuscript (page 11, line 212-230). The fluorescence image of biofilm on the electrode was also provided in Supplementary Fig. 12c in the revised manuscript.

2. As widely reported, many research groups have used the similar ways to achieve record-high interfacial electron transfer efficiency. For instance, Du et al., Environ. Sci. Technol. Lett. 2017, 4 (8), 345-349, intact conductive layer on the individual cell

membrane; Wu X et al., *Angew. Chem. Int. Edit.*, 2011, 50 (2), 427-430, intact conductive layer cross the individual cell membrane. The authors should clearly elaborate the significant improvement of this work.

REPLY: Thanks for this comment. We agree that it is very important to distinguish the current work from the previous publications. We carefully studied these references and other related publications, the main improvements of the current work are concluded as the following:

1. New concept established.

Cell modification is a very hot topic and the frontier of bioelectrochemical research. As mentioned by the reviewer, there are several literatures from different groups published in prestigious journals (e.g., *Angew. Chem. Int. Ed.*, 2011, 50: 427-430; *Nat. Commun.*, 2020, 11:1379). However, all of these work only formed nanoparticles scattering aligned on the cell surface or inside the cell (as shown in the following images copied from these publication), none of them formed interconnected intact conductive layers. Thus, the **electron collection from those individual cell still mainly relied on the bulk electrode**, which was restricted by the 3D architecture of individual cell (Supplementary Fig. S1). On the contrary, the concept established here is the *in-situ* **electron collection from single cell with the “single cell *in-situ* electron collector”**, which is consist of an interconnected intact conductive layer aligned on and across the cell membrane.

Angew. Chem. 2011, Fig. 1c Nat. Commun., 2020, Fig. 4h

The publication from Du et al. (*Environ. Sci. Technol. Lett.* 2017, 4 (8), 345-349) described the modification of the electroactive cell at a biofilm level rather than at individual cell level. Actually, 30 min PDA polymerization (used in the reported work) is hard to form intact layers on individual cell surface as determined in our work.

Thus, this report only formed the protection layer for whole biofilm population as stated by the authors. The main concept of this research is the fabrication of a protection layer for biofilm, which is also totally different from our “single cell *in-situ* electron collector”. Even individual cell can be encapsulated by the PDA, it is only formed nanoshell on the surface of cell membrane, which is totally different from our “interconnected intact on and across membrane” “single cell *in-situ* electron collector”.

2. New record achieved

For MFC performance, the SP collector makes the *S. oneidensis* MR-1 achieved the maximum power output of $\sim 3.18 \text{ W/m}^2$. To date, it is the highest record for this strain under similar conditions (Supplementary Table 1). The carbon dot feeding strategy promoted the maximum power density of *S. oneidensis* MR-1 to 0.49 W/m^2 (Nat. Commun., 2020, 11:1379), which is much lower than our work. Moreover, the *in-situ* electron collector also greatly promoted the electron recovery efficiency to $\sim 86.9\%$, which is also the highest record for the model electroactive bacteria of *S. oneidensis* MR-1. In conclusion, the BES performed achieved here was the new record for *S. oneidensis* MR-1 and was much higher than previous reports.

ACTIONS: We introduced these publication in the revised manuscript (page 11, line 212-230; page 6, line 109-110) and amended a supplementary figure (Supplementary Fig. 1) to clearly distinguish them with the current work. After this revision, we found that the novelty and significance of this work is more clear to the readers. We highly appreciate the reviewer for this comment and suggestion.

3. Electron shuttle is well known to efficiently mediate microbial EET for *Shewanella oneidensis* MR-1. This point should be added into the introduction. Also, more discussion about this issue should also be added. Was this intact conductive layer a type of solid electron shuttle?

REPLY: Thanks for this suggestion. We agreed that electron shuttle mediated EET is

important and should be discussed in this manuscript.

Recently, it was reported that the diffusive electron shuttle (flavins) might only have very minor contribution to the catalytic EET current of *S. oneidensis* MR-1 (PNAS, 2013, 110: 7856-7861; Electrochimica Acta, 2016, 198: 49-55). As the current work mainly focused on the wiring up of electron transfer pathways of redox proteins (it was proved by using comprehensive electrochemical analysis and genetic analysis), we introduced the electron shuttle mediated EET in the discussion part and also provided more discussion as suggested (page 14-15, line 293-316).

For the last question, the protein-FeS or PDA-electrode interaction might be more complex than pure electrochemistry of nanomaterials, which apparently required more in-situ electrochemical and spectroscopic analyses. So, it is premature to conclude it is a type of solid electron shuttle for the conductive layer. As it is out of the scope of the current proof-of-concept research, we deleted previous arbitrary discussion about the electron transfer manner for FeS.

ACTIONS: We added the introduction of electron shuttles mediated electron transfer in the discussion part and more discussions about this issue are amended (page 14-15, line 293-316). Previous arbitrary discussion about the electron transfer manner for FeS was deleted.

4. In order to exclude other reasons to improve the interfacial electron transfer efficiency, the interfacial electron transfer efficiency between the native *Shewanella oneidensis* MR-1 and the electrode modified with PDA or FeS₂ should be examined.

REPLY: Thanks for this important suggestion. To exclude the effect of other factors on the interfacial electron transfer, we presented more control experimental data as suggested in Supplementary Fig. 10. These results excluded other reasons to improve the interfacial electron transfer efficiency and supported that the improvement of interfacial electron transfer was mainly ascribed to the single cell *in-situ* electron collector.

ACTIONS: More experimental data (Supplementary Fig. 10) were amended and more discussion were presented in the revised manuscript (page 12; line 236-241).

5. Considering that an intact conductive layer was on and cross the individual cell membrane, could the interfacial electron transfer efficiency maintain at a high level during the bacterial cultivation period?

REPLY: Thanks for this good question. We agreed that maintain high performance under cultivation is the biggest challenge to the research area of bio-nano-hybrid systems (Accounts of Chemical Research 2016, 49, 792-800; Nature Nanotechnology, 2018, 13: 890-899). Thus, we provided more experiment, more discussions and perspectives about this issue.

The main problem associated was the disruption of nanoshell or dilution of the functional nanomaterials during cell division. In this study, we found that the encapsulation of cell with this single cell electron collector could arrest the cell growth to some extent (several hours to over 18 hours) under growth condition (with soluble electron acceptor) (Supplementary Fig. 11), it would be beneficial to maintain high electron transfer efficiency from several to tens of hours. In addition, under electrode respiration condition (in BES, with solid electrode as the sole electron acceptor), the cell growth of electrode biofilm can be arrested for about 1 day with stable high interfacial electron transfer (Supplementary Fig. 12).

But it is still quite challenge to maintain the integrality of the nano-shell for extremely longer period under cultivation. However, with the technology advances, we believe that this issue could be solved in the near future by developing the next generation single cell in-situ electron collector by using advanced dynamic bio-nano-hybrid assembly approaches (e.g., dynamic nanoshell, self-repairing nanoshell) or genetically engineered synthetic cells (cell with genetic engineered conductive protein shell).

ACTIONS: We evaluated the effect of cell modification on cell growth and presented the data in the revised manuscript (Supplementary Figs. 11-12). More discussions about this issue are also presented as suggested (page 16-17, line 336-357).

Reviewer #2

This submitted work attempts to maximise the interfacial electron transfer and bioelectrochemical systems (BES) performance between living electroactive cells and conductive abiotic surfaces. This engineering attempt is termed as wiring to enhance intercellular connection because close surface contact is essential for maximal electrical output. The author justified the work in reference to current methods that lack in improving the cellular properties for efficient electron transfer, but only focusing on the properties of the conductive abiotic surfaces. For electron transfer improvement, the work targeted two parts of the model electroactive organism, *Shewanella oneidensis* MR-1 (1) “idling” transmembrane electron conduits and (2) “dead” periplasmic electron conduits, both of which are not in contact with electrode surface. To engage these inactive electron conduits, the electroactive bacteria were encapsulated with either polydopamine (PDA) or anchoring FeS nanoparticles in the periplasmic and on outer membrane surface, to act as electron collectors that was shown to increase the BES performance by 1.4- to 3.2-fold, respectively, in comparison to the intact cells. This work demonstrates a proof-of-concept for optimizing BES performance by modifying cell surface chemistry towards contact enhancement with a conductive surface, of which is an attractive strategy to be exploited for potential bioelectronic applications. This is a very interesting paper and a good read. Overall, the results are quite impressive but there are some concerns that need to be resolved:

REPLY: We highly appreciate the reviewer’s positive evaluation of this work.

1. Introduction of PDA and FeS as nanomaterials were rather abrupt. How did the author decide on the materials to be use as electron collectors? Was there a panel of materials compared or at least considered to give the desired pooling of electrons like PDA?

REPLY: Thanks for this reminding. We carefully described the selection of the material and listed a panel of materials compared as suggested in the revised

manuscript (page 6, line 105-110; page 7, line 122-126).

ACTIONS: More details were presented to explain the rationality for PDA and FeS selection (page 6, line 105-110; page 7, line 122-126).

2. In reference to Figs 2 and 3, the TEM image of native SW surface (2d) displayed homogenous distribution of its cytoplasmic content. However, the TEM images of SW@S (2e & 2f) and SW@SP (3c) displayed significant clusters of its cytoplasmic content. It seems to concentrate at the outer membrane in SW@S, while its clumps in SW@SP. Is there any explanation on these observations?

REPLY: Thanks for careful review. I am so sorry for this confusion caused. We carefully checked the problem and found that it was caused by TEM sample preparation. The morphology of the cytoplasm in TEM image was largely affected by the fixation procedure.

1. We found that most of the native SW cells in TEM observation showed cytoplasm morphology (pls refer to the following TEM image, right) similar to that of SW@SP showed in Fig.3. As we did not realize of this difference in cytoplasm morphology previously, the unprofessional selection of the image leded the problem associated the difference between native SW and SW@SP. We now selected the most representative image of native SW cell for presentation.
2. But, as PDA can react with the commonly used fixation reagent of glutaraldehyde, it is difficult to preserve the homogenous morphology of the cytoplasm by using glutaraldehyde, which resulted in the image seems with concentrated cytoplasm around the membrane. So, we tried other fixation procedures. Finally, we observed nearly homogenous cytoplasm matrix with the TEM (pls refer to the following TEM image, left) by using glutaraldehyde (2.5% (v/v)) with paraformaldehyde (2% (v/v)) as the fixation reagent. However, as different fixation reagent used, it is still showed difference with that of native SW cell. According to these results, we can conclude that the morphology of the cytoplasm

in TEM is largely affected by the sample preparation procedure. However, in the current study, we mainly focused on the membrane morphology rather than the cytoplasm. So, to avoid further confusion or misleading, we just highlighted the membrane morphology of SW cell and SW@S cell instead in Fig. 2. With this modification, the reader could focus on the main point of this research (membrane modification) to avoid possible confusion. We really appreciate this suggestion, which helped to improve the manuscript quality and avoid possible misleading.

ACTIONS: New sample preparation method was applied for SW@S cell (method section, page 20, line 408-410). New SEM images (Fig. 2c) was presented to illustrate the whole membrane morphology more clearly and TEM images highlighted the membrane section were presented in Fig. 2d-e.

3. The authors must briefly describe the Live/dead staining method, for example, how long it was incubated post staining, dyes used and excitation/emission wavelengths of the respective dyes. The live/dead stained images, (Fig 2g and Fig 3f), shows all cells to be live. Just wondering how come there is not even a single dead/membrane compromised cell. It would be good to quantify the amount of live/dead cells, post encapsulation (S and SP), from multiple images and present the percentage of live and dead cells.

REPLY: Thanks for careful review. We added the method description in the supporting information. For live/dead stained images, as the cell viability is very high (>98%), it is hard to find the dead cell in a single image at high magnification. To

avoid misleading, we presented the images with lower magnification that can show the dead cells in the revised manuscript (Fig. 2f and Fig. 3f). In addition, we quantified the amount of live/dead cells from multiple images and presented the percentage as suggested in the revised manuscript (page 6, line 120; page 7, line 142).

ACTIONS: Added LIVE/DEAD staining method in supporting information; New images were presented in Fig.2g and Fig. 3f. The percentage of the living cells was also presented in the revised manuscript (page 6, line 120; page 7, line 142).

4. Live/Dead stains gave a reasonable idea on their viability but do these cell surface modifications have any effect on their growth? The surface modifications are not of genetic engineering nature. Hence, it is not a feature that can be inherited for the following generations. Encapsulation/FeS anchoring would be diluted while cells grow. This aspect needs to be explained/discussed.

REPLY: Thanks for this good question. We agree with the reviewer that the surface modification might be disrupted during cell division and the single cell electron collector would be diluted while cells grow.

In this study, we found that the encapsulation of cell with this single cell electron collector could arrest the cell growth to some extent (several hours to over 18 hours) under growth condition (with soluble electron acceptor), it would be beneficial to maintain high electron transfer efficiency from several to tens of hours (Supplementary Fig. 11). In addition, under electrode respiration condition (in BES, with solid electrode as the sole electron acceptor), the cell growth can be arrested for about 1 day with stable high interfacial electron transfer (Supplementary Fig. 12). Thus, this single cell *in-situ* electron collector might be advantage for short- to mid-term application (hours to a day depends on cultivation conditions) as it might provide more stable biointerface without cell growth/death disturbance.

But it is still quite challenge to maintain the integrality of the nano-shell for extremely longer period under growth conditions. However, with the technology

advances, we believe that this issue could be solved in the near future by developing the next generation single cell in-situ electron collector by using advanced dynamic bio-nano-hybrid assembly approaches (e.g., dynamic nanoshell, self-repairing nanoshell) or genetically engineered synthetic cells (cell with genetic engineered conductive protein shell).

ACTIONS: Cell growth curves were presented (Supplementary Fig. 11). Biofilm cell growth and stability of current output were presented (Supplementary Fig. 12). More discussions about this issue were also presented as suggested (page 16-17, line 336-357).

5. The work discussed on the possibility of networking as an electroconductive biofilm. Technically, biofilms form through their embodiment in a self-secreted biological matrix (extracellular polymeric substances, EPS). In relation to my concern #4, if the surface modification is not inherited, the mentioned biofilm could possibly consist of both surface modified and native cells as well as large amount of EPS. In this case, the discussed “construction of artificial conductive network” (page 13, line 252) may still be possible but dampened with presence of unmodified cells and EPS. It would be good if the authors could provide some biofilm (including EPS) images of S or SP capsulated cells. The effect of EPS should be discussed.

REPLY: Thanks for this comment and suggestions. We agreed that production of EPS might dampen the biofilm conductivity and discussed this issue in the revised manuscript. We also presented the biofilm image as suggested (Supplementary Fig. 12c). According to the results, it was found that cell growth could be arrested to some extends in BES (no significant cell growth before 48 h) and no significant EPS could be observed during this period. However, after that, cell growth and EPS could be observed, which would dampen the biofilm conductivity and the interfacial electron transfer.

ACTIONS: Supplementary Figure was presented (Supplementary Fig. 12c) and more discussion about EPS and cell growth was provided as suggested (page 17, line 349-353).

6. The process of S collector assembly has been explained on page 14 (line 289- 294). Process needs more details for the reproducibility. How much time was required for polydopamine coating on the surface of bacteria? How thick is the PDA layer and how does the thickness affect the EET performance? Similarly, since S collector and SP collector are based on nanostructures, how does the size of these nanostructures affect the EET performance?

REPLY: Thanks for these questions. The details for S collector assembly were presented in the revised manuscript accordingly (page 18-19, line 382-389). The time required for S collector assembly is 3 hours and the thickness for PDA layer is about 20-80 nm, which were presented in the revised manuscript accordingly (page 6, line 113-114).

We agree that the thickness and particle size of the electron collector layer would affect the EET performance. Further optimization would be beneficial to improve the EET performance. But, it is now still quite challenge to finely control the thickness and particle size of the electron collector layer without detrimental effect on other important parameters for single cell EET. Comprehensive consideration of the cell viability, conductivity of the electron collector, location on/across membrane, substrate diffusion together with the layer thickness and particle size is required. It is obviously not an easy task calling for systematic optimization and out of the focus of the current work. However, we discussed this issue and provide the perspective in the revised manuscript (page 16, line 331-335).

Instead, we performed optimization for important parameters of single cell electron collector assembly, i.e., polymerization time for S layer assembly (Supplementary Fig. 2b-c) and Fe/S ratio for SP layer assembly (Supplementary Fig. 4).

ACTIONS: Details for S collector assembly were amended in the experiment section as well as the results section (page 18-19, line 382-389; page 6, line 113-114). More discussion on the optimization was also presented (page 16, line 331-335). The optimization for electron collector assembly was also amended (Supplementary Fig. 2b-c; Supplementary Fig. 4).

7. The effect of PCP on the current output has been shown on Page 9 (line 181-187). However, authors did not mention the concentration of PCP used in the methods section (page 17, line 338-340). The concentration of inhibitor is critical as we do not know if the same concentration of inhibitor was used for all 3 types of cells or not.

REPLY: Thanks for careful review. We agree that the concentration of inhibitor is critical. We added the same concentration (2 mg/L) of PCP for all 3 types of cells (native SW, SW@SP, *AmtrC/omcA*@SP). We added this important information in the revised manuscript and also evaluated the cell growth inhibition of PCP. The result showed that low concentration of PCP (2 mg/L) has negligible effect on cell growth.

ACTIONS: The concentration information was added in the main text (page 10, line 200), figure caption of Fig. 5 and Supplementary Fig. 6. The data for effect of PCP on cell growth was also presented in Supplementary Fig. 6b.

8. Bioelectronic setup (page 16, line 322-333) needs more details. The authors mentioned that in-situ assembly of SP collector was done by adding precursors in same ratio (line 328-330). However, the time required for in-situ assembly should also be mentioned here. Was it the same as that required for general process of SP collector assembly on the cells?

REPLY: Thanks. The assembly of SP collector in MFC is slightly different from the general process of SP collector assembly. Due to different medium and different incubation temperature, the time for assembly in MFC (15 hours) is slightly different

from that used for general process (12 hours). We described the details in the revised manuscript as suggested.

ACTIONS: Detailed description was amended in this revised manuscript (page 20-21, line 423-432).

9. It was mentioned earlier (page 15, line 296) that anaerobic conditions are necessary for FeS based SP collector assembly. How anaerobic conditions were maintained during in-situ assembly (page 16, line 322-333)?

REPLY: Thanks for reminding. It is very important to maintain anaerobic conditions for SP collector assembly. For *in-situ* assembly of SP collector in MFC, the anodic medium was firstly purged with N₂ gas to remove dissolved oxygen, the MFC assembly and precursor addition were performed in anaerobic workstation, and then the MFC was tightly sealed to maintain the anaerobic condition during operation.

ACTIONS: More detailed description was amended in the revised manuscript (page 20-21, line 423-432).

10. Figure legend 2g and 3f: Please change “strained” to “stained”.

REPLY: Thanks for careful correction. We corrected it accordingly in the revised manuscript.

11. Supplementary Figure S2: HRTEM image of FeS nanoparticles collected from SW@SP, not clear. Please replace the image.

REPLY: Thanks. We replaced the image as suggested. Pls refer to the new Supplementary Fig. 3c, which is more clear to the readers.

Other revisions made by the authors:

1. Thoroughly grammatical and spelling revision was made.

2. Minor modification on author contribution.
3. Minor modification on Acknowledgement section.
4. Reference update was made.

REVIEWERS' COMMENTS:

Reviewer #1 (Remarks to the Author):

The reviewer is satisfied with the revisions made by the authors. Thus, publication of this revised manuscript is recommended after polishing its English expression.

Reviewer #2 (Remarks to the Author):

The authors have addressed all my concerns reasonably well. I do not have further comments. Congratulations to the authors for the nice piece of contribution.

Authors' Responses to the Reviewers' Comments

(Manuscript ID: NCOMMS-19-42215A)

Reviewer #1 (Remarks to the Author):

The reviewer is satisfied with the revisions made by the authors. Thus, publication of this revised manuscript is recommended after polishing its English expression.

REPLY: We highly appreciate the reviewer's positive comments on this work. According to the comment, we carefully polished the English by inviting a language editing team from SpringerNature. After the polishing, we found the expression of the manuscript is greatly improved. Thanks.

Reviewer #2 (Remarks to the Author):

The authors have addressed all my concerns reasonably well. I do not have further comments. Congratulations to the authors for the nice piece of contribution.

REPLY: We really appreciate the reviewer's positive evaluation.

The comments and suggestions from these two reviewers greatly helped us to substantially improve the quality of this manuscript. Many thanks!